# Automated Inorganic Pigment Classification in Plastic Material Using Terahertz Spectroscopy

**DOI:** 10.3390/s21144709

**Published:** 2021-07-09

**Authors:** Andrej Sarjaš, Blaž Pongrac, Dušan Gleich

**Affiliations:** Faculty of Electrical Engineering and Computer Science, University of Maribor, Koroška Cesta 45, 2000 Maribor, Slovenia; blaz.pongrac1@um.si (B.P.); dusan.gleich@um.si (D.G.)

**Keywords:** terahertz spectroscopy, inorganic pigment classification, deep learning, convolutional neural network

## Abstract

This paper presents an automatic classification of plastic material’s inorganic pigment using terahertz spectroscopy and convolutional neural networks (CNN). The plastic materials were placed between the THz transmitter and receiver, and the acquired THz signals were classified using a supervised learning approach. A THz frequency band between 0.1–1.2 THz produced a one-dimensional (1D) vector that is almost impossible to classify directly using supervised learning. This paper proposes a novel pre-processing of 1D THz data that transforms 1D data into 2D data, which are processed efficiently using a convolutional neural network. The proposed pre-processing algorithm consists of four steps: peak detection, envelope extraction, and a down-sampling procedure. The last main step introduces the windowing with spectrum dilatation that reorders 1D data into 2D data that can be considered as an image. The spectrum dilation techniques ensure the classifier’s robustness by suppressing measurement bias, reducing the complexity of the THz dataset with negligible loss of accuracy, and speeding up the network classification. The experimental results showed that the proposed approach achieved high accuracy using a CNN classifier, and outperforms 1D classification of THz data using support vector machine, naive Bayes, and other popular classification algorithms.

## 1. Introduction

The electronic circuit’s limitation and the excessive capability of optical devices to excite terahertz (THz) waves in the past have inhibited the THz spectrum exploitation in various industrial and laboratory applications. THz is electromagnetic radiation between the microwave and the infrared region within the frequency domain of 0.1–10 THz [1]. The THz waves are non-ionizing, non-invasive, and penetrate many materials. In the last two decades, research and development probing the THz spectrum and THz signal generation has led to technologies in several scientific applications [1,2,3]. Nowadays, THz applications are often used in biology, physics, chemistry, material science, and security. Also, it is known that THz radiation is very sensitive to translation, vibration, and rotation [2,3]. Such distinctive characteristics enable using the THz spectrum in numerous inspection, scanning, and imaging applications. The THz frequency spectrum’s uniqueness allows new research to be conducted in the scientific and engineering fields. 

Regarding the overview of the latest scientific publications, most THz applications are limited strictly to scientific research with the minority on engineering applications. The main reason stems from the fact that the engineers’ community is not aware of the technology, and there is a lack of ‘off-the-shelf appliances’. Many THz systems contain expensive elements such as photoconductive antennas, laser sources, low noise, and sensitive amplifiers to acquire the emitted radiation. With continued research on the physical principles of radiation, excitation, and probing, THz applications can be a breakthrough technology in the new upcoming industrial era. They can be used in various industrial applications and can offer valuable upgrades or replacements for standard devices. 

This paper presents an automated inorganic pigments (IP) classification of plastic materials using terahertz frequency domain spectroscopy (THz-FDS) as a source of data. The vast majority of the THz spectroscopy is done with the time domain spectroscopy (TDS) technique, which is robust and flexible. The TDS uses short pulses and coherent detection, which consequently disables standing waves and simplifies the data analysis. The frequency resolution is in the span of 1–5 GHz [4]. Regarding unknown spectral properties of the pigments and the intention of accurate spectral properties examination, the THz-FDS technique was used. The main advantage of THz-FDS is high-resolution specter measurement, possible user-selected frequency span, and are cost efficient. Unlike TDS, FDS resolution is in the range of 1–5 MHz. Due to the continuous waves operation principle and possible standing waves occurrence, the FDS data are more complex and require advanced post-processing techniques [4,5]. The analysis of FDS data is discussed in depth below.

Plastic materials are used widely in many fabrication processes, assembly, and design. The pigments’ classification of the processed workpiece is essential in quality control, material characterization, and fabrication validation [6,7]. The production of raw pigmented plastic material is a trade secret, and the manufacturer’s confidential information is usually not accessible to the client. To ensure supervision over material characteristics, the demand for non-invasive and non-destructive material inspection arises in many production assemblies. Currently, the inspection techniques used for plastic materials in industrial environments are mostly destructive, including cutting and grinding the sample. On the other hand, non-invasive techniques are based mostly on surface inspection methods [7]. For example, the classification of the colored material can be achieved straightforwardly with machine vision, which is inefficient for in-depth analysis [8]. Also, inspections based on thermal, microwave, ultrasonic, acoustic emission, and magnetic techniques are not adequate for IP classification [1,9]. The inability to use well-established inspection techniques for non-destructive testing of plastic materials has THz technology as its advantages [10]. THz waves can penetrate the polymers, even if the material is opaque and has a high spatial resolution compared to microwaves [11]. Thus, THz technology is suitable for contact-free inspection and can be used for crack, defect [12], mechanical stress [13], and aging process detection [14] in polymer materials. Most inspections were done with the TDS and some with the FDS with know processing techniques, such as phase fringes extraction and FFT [3]. All applications and their data analysis were made for the characterization of a single instance of the material. Unlike the aforementioned, this paper introduces a high-frequency resolution approach with advanced data processing of complex FDS data, which enables pigment classification and the possibility of automated quality control of the plastic material. The presented work extends THz spectroscopy as a real-time system for non-destructive IP classification and validation.

Most of the data analysis in THz-FDS is undertaken by determining the attenuation and the phase shift of THz waves spread through the medium [15]. The detected absorption lines of a medium are measured and compared with the reference values, where complex permeability was estimated using the Kramers–Kronig relation [16]. Besides, chemometric methods, such as principal component analysis [17], individual component analysis [18], and partial least square [19], are used for finding the compounds in the measured medium. The THz-FDS systems based on photoconductive antennas (PCAs) utilize the two optical signals with different wavelengths to create a modulated optical signal with THz frequency [3]. A most common method of measuring the spectral components is by sweeping the modulated THz frequency and measuring phase fringes. The attenuation and phase shift of THz waves in the medium can be estimated from the measured phase fringes. Such a THz-FDS system needs careful calibration and setting. The calibration’s first concern, which needs to be examined, is the measurement’s uncertainty due to the tuned distributed feedback laser diodes (DFLD) used in THz-FDS systems. When sampling the detected THz wave, the measurement’s actual frequency can vary regarding the reference frequency. The second main calibration task is the environmental impact. THz waves are sensitive to the change of temperature and change of humidity around the measured sample. Therefore, all measurements are usually performed in a controlled environment, which is a significant limitation for industrial applications. The authors in [16] predicted that THz technology would be used more in laboratory environments than the industrial production processes. Material analysis in quality control and waste management could be areas where THz technology would have an edge. According to this, THz waves are not ionized waves; therefore, they would not damage the measured sample. Many materials have spectral footprints in the THz band. THz technology could also detect unwanted metals in the measured medium, since metals are opaque to the THz waves. There are some examples in material and natural process control, such as fruit inspection [20] and fermentation supervision [21]. The major drawback remains that the applications are performed in a controlled environment and not in real-time.

By reviewing the literature, we found that the classification of materials using a THz sensor is extremely difficult due to the wide THz spectrum, and the measurement can be contaminated with different uncertainties, such as DFLD non-linearities, measurement bias, and environmental impact. All the influences are known, but cannot be determined and compensated precisely in the measurement. The supervised learning procedure of the machine learning (ML) algorithm can be used according to the known indeterminate uncertainties. Supervised learning ensures robust mapping of ML over a set of input-output data pairs. All data pairs contain uncertainty, which can be suppressed efficiently under the assumption that the significant futures are preserved and hidden. Many different algorithms of ML exist, which can be deployed for IP classification. After extensive research and comparison between different ML strategies, the convolution neural network (CNN) gives beneficial results. The CNN algorithm is a subset of deep neural networks and deep learning paradigms [22], and has proven its effectiveness as an image, speech recognition, face detection, futures extraction algorithm. The novel research confirms that CNNs have advantages in series forecasting, a data-driven approach for diagnostic and fault classification of various industrial processes and applications [23,24,25,26,27].

The paper proposes an automated IP classification using a novel preprocessing algorithm suitable for processing with CNN classification methods [22], which can be executed in real-time. The acquired THz signal was transformed from 1D to 2D representation and classified using a CNN. The 1D data are transformed into 2D data representation by preprocessing 1D THz data with peak detection, envelope extraction, and downsampling algorithms, which operate over the THz phase fringes. A new algorithm, called Windowing with Spectrum Dilatation (WSD), transforms 1D data into 2D data that represent a material’s spectral features obtained using the THz-FDS. The 2D data are classified using a CNN, where the material’s spectral futures are distributed spatially throughout the 2D data. Such a transformation ensures that the spectral futures are located regionally with a certainty boundary. Classification and detection of spatially spread futures with added uncertainty is the main advantage of the CNN algorithms. The complexity of CNN is related to the preprocessing parameters’ selection, and can be treated as an optimization procedure. The proposed method for IP classification with CNN was evaluated experimentally. Plastic material, polyethylene (PE), was mixed with various IPs and used as an evaluation sample. The CNN was trained with the preprocessed training set. The efficiency of the training is closely related to the selection of the preprocessing algorithm and its parameters. The paper compares novel WSD and the mostly used set cut-technique (SetCT). The advantage of WSD over SetCT was confirmed with the experimental results, and the outperformance is evident. The proposed approach WSD-CNN was also compared with other known classification algorithms, such as support vector machine (SVM) [28], naive Bayes (NB) [29], classification tree (CT) [30], and discriminant analysis (DA) [29], all of which operate over 1D data. All the preprocessing methods are discussed and compared later in the article. The spectral characteristic of each PE sample was gathered between 0.1 THz and 1.2 THz. As we confirmed during the work, the proposed WSD preprocessing for automated IP classification based on THz-FDS with the CNN classification achieves high reliability, robustness, accuracy and can operate in real time. The paper also shows the improvement of the THz-FDS scanning technique with efficient and robust complex FDS data analysis.

The paper has six sections. The Section 1 is an introduction, where the main objectives of the work are presented. The Section 2 represents the THz-FDS operation principle and experimental setup with the inspection samples. The Section 3 introduces different preprocessing algorithms of the THz-FDS measurements. The following section continues with CNN structure selection, hyperparameters’ role, and the benefits of the WSD preprocessing algorithm. The experimental results are presented in Section 5, where comparisons are conducted between different preprocessing and CNN structures. The paper is concluded with Section 6

## 2. Terahertz Frequency Domain Spectroscopy Principle for Inorganic Pigments (IP) Classification

The TeraScan 1550 from Toptica Photonics, Munich, Germany, was used for the THz-FDS experiment. The Tarascan 1550 can generate THz waves in the span of 0.03–1.21 THz. It has high THz power and a wide dynamic range. It utilizes mixing or beating the optical signals excited from the two DFB laser diodes with different wavelengths and a PCA emitter and detector. The experimental setup is presented in Figure 1.

The experiment involved five different categories of samples with individual inorganic pigments, which differed in five colors, white, blue, green, yellow, and black. The geometrical parameters of the samples were equal in all the used batches. The samples with different pigments are presented in Figure 2.

The sample organization for the deep learning algorithm will be discussed in section five. Only the essential operation principle of the Terascan 1550 is presented for a better overview of the approach and the general understanding of the readers. The system comprises two independent laser sources for signal modulation and two PCAs for emitting and detecting THz radiation, depicted in Figure 3.

Two tunable DFB lasers were utilized for generating optical signals with different wavelengths, $\lambda_{1}$ and $\lambda_{2}$, and mixed within the optical fiber coupler. The resulting optical signal was modulated with frequency f, which can be expressed as,
(1)$$f=c\cdot n\cdot \frac{\Delta \lambda }{\lambda_{1}\lambda_{2}}$$
where c is the speed of light in the vacuum, n is effective refractive Index (n≈1.4682 for an optical fiber) and Δλ is the wavelength difference, given as, $\Delta \lambda =\lambda_{1}-\lambda_{2}$. The two DFB laser diodes were of the same type and emitted a light’s wavelength of one DFB. Thelaser diode is shifted by cooling and the others by heating [31,32].

The optical source was coupled with the PCAs. The PCA emitter acts as a capacitor with the charge $E_{Bias}$ if the antenna gap is not lit [33]. The photocurrent is induced when the gap is lit with the optical signal. The photocurrent drives a dipole antenna, and the THz wave is established [34]. The THz far-field $E_{THz}$ is estimated as,
(2)$$E_{THz}\left(r,t\right)=-\frac{A}{4\pi r\varepsilon_{0}c^{2}}\frac{d}{dt}\cdot J_{s}\left(t\right)=E_{0}\left(r\right)\cos\left(2\pi f\cdot t+\phi \right)$$
where A is the light illumination area, r is the distance from the source, $\varepsilon_{0}$ is the dielectric constant, $J_{s}\left(t\right)$ is the induced surface current in the PCA gap, and $E_{0}$ is the distance dependable peak value. The detector PCA acts in a similar way to the emitter PCA, where the THz waves pushed the photocarriers apart with induced voltage $V_{THz}=V_{0}\cdot \cos\left(2\pi f\cdot t+\phi_{THz}\right)$.

In a transmission-based FDS system, such as that shown in Figure 3, the emitter PCA’s emitted beam is collimated through the sample into a PCA detector. The measured characteristic of the material is transmittance. The transmittance can generally be described as how much of the emitted field has passed through the measured media. Transmittance T is defined as,
(3)$$T=\frac{n_{2}{\left|E_{T}\right|}^{2}}{n_{1}{\left|E_{I}\right|}^{2}}\approx \frac{I_{T}}{I_{0}}$$
where $E_{T}$ is the remaining field after propagation through the medium, and $E_{I}$ is the emitted field,$\;I_{T}$*,* $I_{0}$ are the measured and initial intensity, respectively. Regarding the classical electromagnetic theory using the Maxwell equations [1], the THz wave propagation can be described accurately. The focus is on the frequency-dependence on absorption and dispersion in the measured medium with the transmission spectrometer. Absorption affects the amplitude of the propagated wave, and is described with the absorption coefficient α(λ). The attenuated intensity of the propagated wave is described as,
(4)$$I_{T}\left(d,\lambda \right)=I_{0}e^{-\alpha \left(\lambda \right)d}$$
where d is the propagation depth or thickness of the inspected material, and λ is the radiation wavelength. The dispersion or change in propagation speed will cause the propagated wave’s phase change, as shown in Figure 4.

The measured photocurrent in the detector PCA’s gap will depend on the emitted frequency and the absorption and dispersion in the medium. The THz far-field will also drop with the square of the distance between the emitter and detector PCA. Nevertheless, the distance between the PCA emitter and detector should be considered a phase shift, as shown in Figure 5.

In the presented setup shown in Figure 1, a fixed distance is considered between the PCAs. For measuring an attenuation and phase shift in a medium, the frequency should be swept. With sweeping, the frequency f induced photocurrent obtains a sinusoidal form (phase fringes) due to the interferometry between the optical signal and THz waves in the PCA detector. An example of the measured phase fringes of the sample in Figure 2 is shown in Figure 6.

The amplitude and phase can be extracted from the measured phase fringes in Figure 6. The extracted amplitude and phase, with advanced processing algorithms, can be used for automated material classification and non-invasive inspection. The data processing for automated IP classification is followed up in the next section.

## 3. THz Data Processing

THz spectroscopy creates an immense amount of data which need to be processed. The experiment’s main idea and the presented methodology would confirm that the IP components mitigate and affect the THz microwaves in the unique futures. All the spectral futures in the measurement are surreptitious, and are not expressed clearly. The theory can confirm the hypothesis, but automatic classification needs in-depth analysis and the employment of a reliable processing technique.

As mentioned, the acquired spectroscopic data contain profound information about the IP components, the pigment’s color and the prepared pigmented material’s quality. The prepared material’s quality is classified as a dye’s spatial density through the medium and the fabricated material’s quality. The preprocessing algorithm needs to be employed before the efficient classification with the CNN proceeds.

### 3.1. Preprocessing the Measured Phase Fringes

The preprocessing of the acquired data involves the transformation of the series in a 2D single-channel image. The series can be transformed in different ways. The most common way to convert the sequence to a 2D array is presented in a data-driven approach (DDA). The DDA is often used for series forecasting [35,36], and its array structuring is appropriate for big data organizations [37]. The DDA implies data acquired at a particular time, and, regarding the classifier algorithm, forecasts future values and events. Often it is applied for fault diagnostics based on a large amount of data from different sources [38].

Regarding the DDA methodology, the acquired THz data can be organized similarly. The main drawback of DDA with THz-data series is the significant rise of the computational burden, which demands a complex CNN structure with an extended processing and learning phase. The DDA does not ensure sufficient robustness to the measurement uncertainty, which limits classification efficiency and reliability significantly. The 2D transformation procedure, which is based on four steps, is detailed in the presented work. The first three algorithms start with peak detection, envelope extraction, and the sub-sampling procedure. The last step introduced a windowing spectrum dilation (WSD) algorithm, which transforms 1D data into 2D data. The first three algorithms are used for WSD to lower the complexity of the data and the CNN structure. The proposed WSD is crucial for further classification and computational complexity. All four steps introduce various setting parameters, which affect the CNN’s efficiency. It is important to mention that there are many different possible approaches to transform a series into an array. The paper describes the unique procedure for THz data, which gives more expedient results. All four steps are discussed hereafter.

### 3.2. Peak Detection, Envelope Extraction, and Downsampling

The first two steps of preprocessing are peak detection and envelope extraction. Envelope extraction is based on quadratic spline interpolation over peak values. The peak-find function (PFF) objective is to find the local maxima in raw measured data, with two main parameters: Threshold value and the peak neighborhood interval. The threshold value ensures extraction of the upper envelope and rejects small extremes, whereas the peak neighborhood reduces lower near lying extremes. The quadratic spline interpolation follows after the PFF processing. The quadratic spline interpolation is used to preserve the correct envelope shape to cover all significant envelop transitions. The spline interpolation is calculated regarding the piecewise splines’ selection with given measured knots. Each interval length is selected arbitrarily and is an accuracy concern of the processed envelope. The interval length is fixed for all measurements. The last step before WSD is the downsampling algorithm. The downsampling procedure reduces the data complexity, and will be discussed in the section with WSD.

### 3.3. Data Series Transformation with the Windowing Spectrum Dilation Algorithm

The last step to prepare THz data for CNN processing is transforming the downsampled envelope to a 2D image. Before the WSD is discussed, the set cut-technique (SetCT) will be presented briefly. SetCT is often used in DDA, where the data series is distributed equidistantly in the matrix column or row. The technique performs well on the data set, where the normal distribution and standard deviation are slowly varying or constant. The SetCT transformation for the image ISetCT∈ℝm×n, m∈ℕ,n∈ℕ over data set D∈ℝ1×r*,*
r∈ℕ can be described as,
(5)ri=Din=|dm|Di={x|x∈{Dj}j∈ℕ,(1+(i−1)n)≤j<(1+i·n)}ISetCT={ri|ri∈D, 1≤i≤m}
where Di⊂D*,*
Di∈ℝ1×n*,* d∈ℕ*,*
ri is the i−th row of the image, i-is the running index on the interval i=[1,2,…,m]. For the presented THz data set, a similar approach does not provide adequate results. The main issues are measurement uncertainty and the inability to align the measured spectrum accurately with the reference data set. All the uncertainties are reflected in the spectrum shift, amplitude leak and asymmetry of the previous acquisitions. A listed effect prevents accurate data spectrum alignment and reduces the efficiency of the SetCT technique. The SetCT technique is presented graphically in Figure 7.

The WSD method is presented due to the given facts. The WSD transforms the series so that each image row contains data from the beginning to the dilation factor. Each image row extended the spectrum stepwise with a dilation parameter. At the end of each consecutive row the original data belong to the higher measured specter. It follows that each subsequent row contains the data of the dilated specter, where the last row embraces the whole specter. The proposed approach reduces measurement uncertainty, and overcomes the problem with spectrum alignment. The WSD transformation for image IWSD∈ℝm×n, m∈ℕ,n∈ℕ for dataset D is described as,
(6)ri=Din=|dm|Di={x|x∈{Dj}j∈ℕ,j=i·k}IWSD={ri|ri∈D, 1≤i≤m}
where ri is an i-th row of the image, i—is a running index on the interval i=[1,2,…,m] and k is a dilation factor. Dilation factor k is determined as k=s·n!, k∈ℕ, s∈ℕ, where s is a dilation stride and can be selected arbitrarily. The WSD is depicted in Figure 8.

The WSD transformation can be used on different data sets and preprocessing stages. Regarding the downsampling algorithm and WSD, the approach is practical for lowering image resolution while still preserving data characteristics. From the WSD adjacent image row’s observation, each further row contains the data set, which is extended in the measured spectrum, but downsampled regarding the data set. In other words, each new layer of WSD image represented the low-pass values of the data. This means that the adequately downsampled data preserves mainly the shape of the envelope in all image layers. The aforementioned confirms that the WSD image transformation is, consequently, more resilient to the uncertainties. The image does not contain sharp edges at the first rows of the image, as can be noticed in the SetCT method. Sharp edges are mainly a consequence of the image assembly with a fixed length, where the extracted envelope is decreasing monotonically. The preprocessing algorithm contains PFF, envelope extraction, the downsampling algorithm, and WSD transformation. The sequence of the preprocessing algorithms is presented in Figure 9.

## 4. Convolutional Neural Network Structure Selection and Learning Procedure

The CNN is used to classify different IP content in the plastic material. The significant advantage of CNN is the recognition’s robustness of spatially distributed futures, which mimics the eye’s natural phenomena of cortex vision [35,36,37,38]. The pigment classification in plastic material using the THz-FDS data is not well researched regarding the overview of the scientific works. The characteristic of the THz-FDS data with futures’ dispersal in the different frequency domain for each IP component and uncertainties makes classification complex, whereby known classification algorithms’ schemes provide poor results. The THz-FDS data need additional analysis, and most preprocessing methods and standard classification structures are inefficient. As is confirmed in section five, the classification with 1D THz-FDS data and algorithms such as SVM, NB, CT, and DA, does not achieve sufficient accuracy and reliability. Therefore, the 1D data are transformed to 2D with WSD, and the CNN classification procedure is used. CNN’s main advantage is that it does not need an additional algorithm for locating the future in the 2D data set. The future can be extracted, even if it is randomly spatially distributed, which is beneficial regarding the THz-FDS data characteristics. With the proper selection of the WSD and CNN parameters, the accuracy of classification is evident. The efficiency of the CNN depends on the prepared training data set, network structure, and hyperparameter selection [39,40], which is discussed later in the article.

### 4.1. Convolutional Neural Network (CNN) Structure and Hyperparameters’ Selection

Regarding the preprocessing algorithms from Section 3, the CNN structure is selected for two types of the 2D data set. The first CNN structure is tuned for a high-resolution and the second CNN for a low-resolution data set. The objective of the parameters’ tuning is to design a CNN classification structure which satisfies the real-time criteria, accuracy, and reliability. The selection of CNN structure and layers depends on the complexity of the 2D data set preprocessed with SetCT or WSD. The depth of the network is defined by the number of consecutive layers, where each layer is related to the ability of the network to learn specified complex patterns, which is desirable for IP classification with hidden spectral futures. With each added layer, the network gains additional leeway for knowledge complexity. On the other hand, too many layers can lead to overfits. The overfit impairs the network’s accuracy and reliability to operate with samples which are not associated directly with the training set. The number of kernels, size and stride needs to be selected in the design of the CNN structure. The CNN parameters’ selection and validation is a complex optimization procedure that is extremely time-consuming and requires many iterations and analyses [26]. Four preprocessing methods are used in the presented work. All four methods use envelope extraction, and transform 1D data in high and low-resolution 2D data with SetCT and WSD. High-resolution 2D data use more complex CNN, where low-resolution use a simplified CNN with fewer layers and hyperparameters. Figure 10 and Figure 11 present the CNN structure for full and low-resolution 2D data with SetCT and WSD transformation.

In Figure 10 and Figure 11, Conv, Relu, Maxpool, FC, and Cross are abbreviated for convolution operation, rectified linear activation function, max pooling, fully connected layer, and cross-entropy, respectively. The numbers in Figure 10 and Figure 11 after the forward slash represent the layer stride.

The CNN structure starts with the normalization function, which scales the input 2D data to an arbitrary span. Linear unit normalization was used in our work. The first layer starts with convolutional filters (CF) or kernels. The CF is a fixed valued moving frame with a preselected structure and stride. The repeated pattern extraction is allowed with the convolution operation over the inspection image with the kernel. If we represent the image as I∈ℝm×n and kernel as k∈ℝp×l, p∈ℕ,l∈ℕ, where p<m and l<n hold the convolution for one channel image is presented as,
(7)yv(i,j)=ψ(∑v=1s(I∗kv)(i,j)+b)
where v is the number of kernels in the individual convolution layer, ψ is an activation function, b is a bias value, and (i,j) is the position of the given pixel. The output of the convolution operation is a future map image yv. With a higher selection of v the capability of future extraction is improved, whereby the network complexity is increased. The CF selection is a tradeoff between network complexity and extraction potential. For the activation functions, the ψ rectified linear activation function is used (ReLu). The ReLu function speeds up the learning procedure, and is not prone to vanishing gradient. The ReLu is described as,
(8)ReLu(x)=max(0,x)
where x is a single pixel of the image yv. The second layer, which is followed mainly by the convolutional layer, is the pooling layer. The pooling layer reduces the image size by extraction of the highest future value in a certain area, determined with a pooling kernel size and stride. The used pooling kernel is presented as,
(9)Maxpool(p)=max(0,Ph(i,j))
where Ph⊂ yv with Ph∈ℝh×hh∈ℕ and mean kernel size. The number of layers is correlated with the classifier’s complexity. The given experiments show that the pooling layer can be omitted with WSD processing, due to the downsampling procedure in the preprocessing phase. This is a clear indication that the WSD can reduce network complexity. The last layer of a CNN is a fully connected layer. The fully connected layer uses the 2D data from the previous layer, and turns it into a single vector, where each neuron is connected to all other neurons in the neighborhood with a given weight. Each neuron calculates the output value as,
(10)yi+1,k=η(∑l=1q−1xi,jwi,j)+bl
where q is the number of neurons in the layer, xi,j and wi,j are input signals with associated weight, i is a position from the previous neuron or index of the flattened vector, j is the current input and bl is the bias of the l-th neuron. The output of the layer is the probability value of the classified set with size z, Y=[y1,y2…,yz]. Regarding Figure 2 and the number of different samples, the output size is equal to z=5.

### 4.2. CNN Training

After the CNN structure selection with different convolution and pooling layers, the training process is associated with hyperparameters’ determination. The CNN training is based on cost function minimization, with appropriate adjustment of the kernel coefficient and fully connected layer weights. The network is trained with an ADAM optimizer, the performance of which has been confirmed by many scholars [41,42,43]. The ADAM algorithm is an extension of the stochastic descent gradient approach, and combines the adaptive gradient and root mean square propagation algorithms [44]. The IP classification’s accuracy and reliability are related strongly to the number and types of layers and the preprocessing algorithm.

Each of the CNN structures presented in Figure 10 and Figure 11 were trained regarding the preprocessed (SetCt or WSD) 2D input data. The raw 1D data size obtained from THz-FDS contained 20,000 samples in the frequency span of 0.1–1.2 Thz. After transformation, full-resolution 2D data had a size of 50 × 400, whereby low-resolution data were 50 × 8. The low-resolution size was achieved during extensive optimization and iteration. The full-resolution and low-resolution 2D data with SeCT and WSD techniques are presented in Figure 12.

The THz-FDS data for CNN training, validation, and testing were acquired from different batches with five different inorganic pigments (see Figure 2). Each batch contained 100 samples of each pigment. For CNN training, 10 batches were used, which meant 1000 samples for each pigment. The samples were divided into training, validation, and testing groups. Each group contained the same number of samples for each pigment. Such an approach offers some guarantee and control over the learning phase to avoid the CNN overfits. The sample group covered 50% of training, 30% of validation, and 20% of test samples, which was 500 samples of each pigment for training, 300 for validation, and 200 for testing. The CNN learning times and achieved results are presented in Table 1. The table’s first column $\mu_{l}$ presents the learning accuracy. The second column is learning time $t_{l}$, given in hours. The third column α is the initial learning rate, and the last is the maximal number of preselected epochs.

## 5. Experimental Results

The classification algorithms were tested on the THz-FDS data acquired with the TeraScan 1550 presented in Figure 1. The TeraScan system was calibrated only for measurements in the first batch. All other batches used the initial system setting, which is not usual for such applications. The THz scanner often requires recalibration, which alleviates the external impact and measurement uncertainty. On the other hand, recalibration is time-consuming and reduces system usage. To avoid excessive recalibration and ensure classification reliability, the robustness property of the algorithm is crucial for real-time scanning. The frequency span of the TeraScan system was set to 0.1–1.2 THz, with 20,000 samples for a single measurement. The peak detection and envelope extraction used the maximum seek procedure in a span of 150 samples, where the downsampling ratio was set to 1/50. The experimental procedure is depicted in Figure 13.

The results in Table 2, Table 3, Table 4, Table 5 and Table 6 and Figure 14 present the comparison between the SetCT and WSD transformation for high- and low-resolution 2D data, with different CNN structures presented in Figure 10 and Figure 11. Table 7 and Figure 14 present the comparison between different classification algorithms based on 1D and 2D input data.

The obtained results of the CNN classification with SetCt and WSD are presented in Table 2, Table 3, Table 4, Table 5 and Table 6. The Table parameters are: $t_{t}\left(s\right)$ is the average processing time for a single item of 2D data, $\mu_{t}$ is the average accuracy,$\;\sigma_{t}$ is the standard deviation of the $\mu_{t}$, and $\mu_{tmax}$, $\mu_{tmin}$ are the best and worst probability values of the testing samples.

The confusion matrix of CNN classification is presented in Figure 14, where rows and columns represent actual and predicted values, respectively.

The comparisons with support vector machine, naive Bayes, classification tree, and discriminant analysis [24,25,26] are presented in Table 7 and Figure 14. All classification algorithms SVM, NB, CT, and DA used 1D data. The preprocessing data included peak detection, envelope extraction, and downsampling algorithms with equal parameters, as in WSD_low_. Table 7 presents the achieved scores, where $\mu_{a}$ is the average accuracy of the complete validation set for each classification algorithm separately.

Figure 15 presents the confusion matrix of the classification algorithms SVM, NB, CT, and DA.

The presented results in Table 2, Table 3, Table 4, Table 5 and Table 6 demonstrate the efficiency of the learning algorithm with the preselected testing group, and show significant differences among the approaches and CNN structures. The WSD method has a beneficial property, and is efficient with the reduced data set. It can be seen that the downsampling ration 1/50 does not influence the classification accuracy much; see Table 2, Table 3, Table 4, Table 5 and Table 6 row $\mu_{t}$. In contrast to the SetCT_low_, such a reduction cannot be applied. The full data set, without reduction, showed promising efficiency of the IP classification, whereby it requires longer training and operation time, Table 2, Table 3, Table 4, Table 5 and Table 6 row $t_{t}$. The selection of the CNN_full_ structure is more demanding and time-consuming than CNN_low_, wherein kernel size, stride selection, and layer sequence require more attention and are more sensitive to parameter change than CNN_low_. Comparing both reduced and full-data sets shows that the WSD is a more suitable technique than SetCT. The advantage of WSD is in the 2D data structuring and CNN convolution layer operation, where the claims discussed in Section 3.3 are confirmed with the experiments. The reduction of the 2D data also benefits the CNN structure. Figure 10 and Figure 11 show that the CNN with the lower 2D data does not contain a max-pooling layer. It was confirmed that WSD has an advantage as a data reduction technique and structure relaxation for the THz-FSD measurement and CNN. A max-pooling layer with WSD_low_ lowers CNN’s reliability, which means that the reduced 2D data after downsampling contained all the significant spectral futures of the measurement, and additional simplification inside the CNN eliminated them. Apparent differences can be seen from the confusion values in Figure 14 by WSD_full_ and WSD_low_, where the scattering of the SetCT scores is evident, which means the downsampling of SetCT deteriorated the results. The WSD confusion values in Table 2 and Table 6 are more consistent than SetCTs and indicate that the WSD approach had higher robustness. The WSD advantage can finally be confirmed with an accuracy of reduced WSD_low_, which is comparable with WSD_full_. Comparing the classification algorithms in Figure 15 and Table 7 shows that the WSD achieved the highest results and outperformed the 1D-based algorithms. Regarding the confusion matrix in Figure 14, DA performed better than the SVM, NB, and CT but still reached lower results than the 2D-based algorithms, except for SetCT_low_. SVM and NB achieved results with an approximate accuracy of 70% and CT close to 50%. It needs to be mentioned that, with the use of unprocessed 1D data, the accuracy of SVM, NB, CT, DA was much lower than with the preprocessed data presented in Table 7. The comparison shows that the approach using WSD-CNN over THz-FDS data had higher accuracy and the best performance. The transformation from 1D into 2D data with WSD was beneficial for the THz-FDS approach. The paper shows that the IP component in the plastic material can be classified with THz-FDS data and the proposed WSD-CNN structure with an average accuracy of 98%. It is also evident that each IP in plastic material has unique spectral characteristics in the THz domain and non-destructive analysis is possible.

## 6. Conclusions

This paper presents an automated inorganic pigment classification in plastic material using THz spectroscopic data. The efficiency of 2D WSD transformation with CNN classification gave promising results, and outperformed similar classification techniques based on 1D input data. The 2D transformation of THz-FDS data showed a strong correlation with the classification accuracy, where the efficiency of WSD was confirmed with the experiment results. The WSD was more robust on uncertainties and bias, and enabled considerably more data simplification with insignificant reduction of the classification reliability. Data reduction speeds up the CNN processing, and can be appropriate for a real-time operation, where the accuracy is not the substantial trade-off between speed and accuracy. The work can be extended to additional pigments, and all the requirements can be preserved.

The current state of the presented work shows additional possibilities for improvement. The first option could be exact determination of the THz span with higher pigment sensitivity. Such an approach will, additionally, reduce the data complexity and speed up classification. The second option would be regarding the WSD technique. The WSD can also be used in the opposite direction, starting at the end of the THz-FDS specter. In this case, the higher frequency data would be exploited notably, which could increase reliability. Finally, a very promising improvement, which should be worthy of investigation is the THz data transformation into the 2-channel 2D data. The two-channel 2D data can represent complex values, where extraction would be needed of the phase from the THz-FDS data.

## Figures and Tables

**Figure 1 sensors-21-04709-f001:**
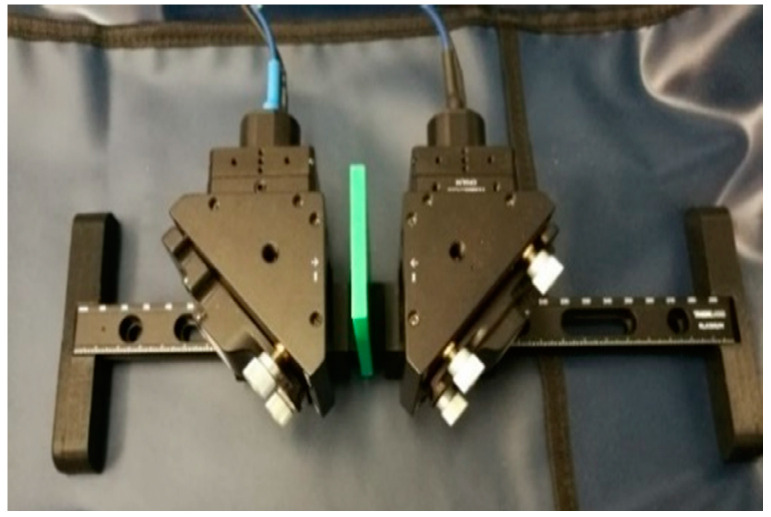
Experimental setup with photoconductive antennas (PCAs) and plastic sample in green.

**Figure 2 sensors-21-04709-f002:**
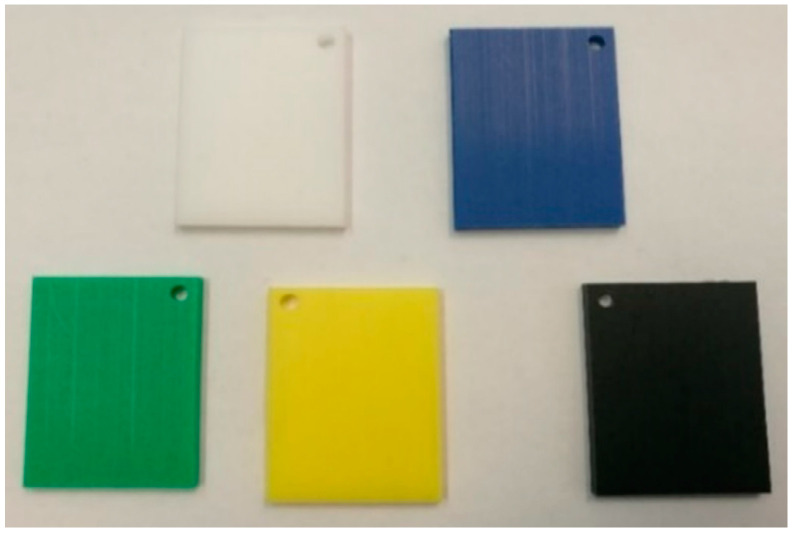
Five plastic samples with different inorganic pigments.

**Figure 3 sensors-21-04709-f003:**
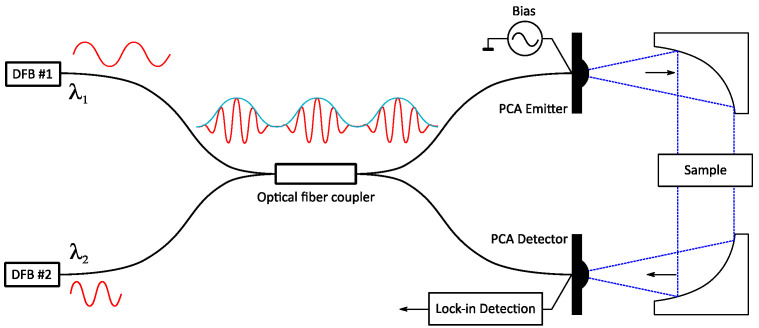
The structure of the Terascan 1550 THz-FDS (frequency domain spectroscopy) system.

**Figure 4 sensors-21-04709-f004:**
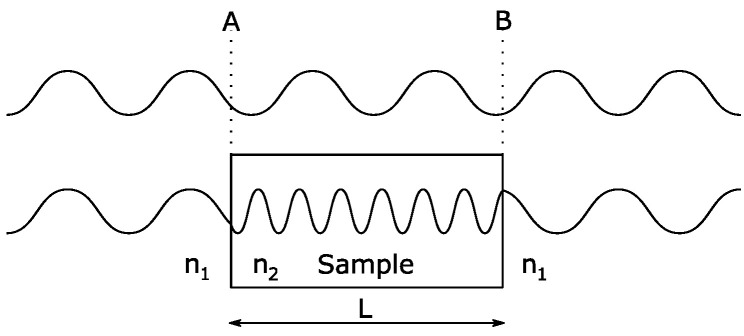
Dispersion in the measured medium.

**Figure 5 sensors-21-04709-f005:**
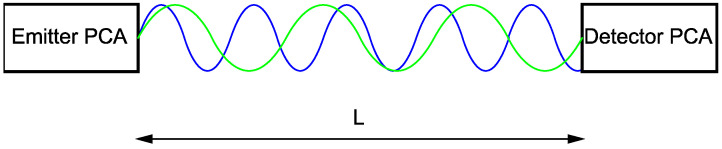
Phase change dependency on the variable frequency at fixed distance *L* between emitter PCA and detector PCA.

**Figure 6 sensors-21-04709-f006:**
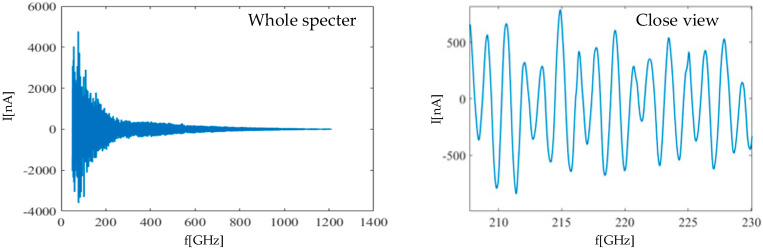
THz frequency sweep-phase fringes.

**Figure 7 sensors-21-04709-f007:**
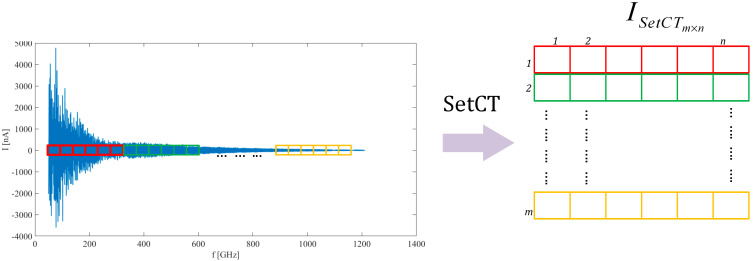
Set cut-technique (SetCT) transformation technique.

**Figure 8 sensors-21-04709-f008:**
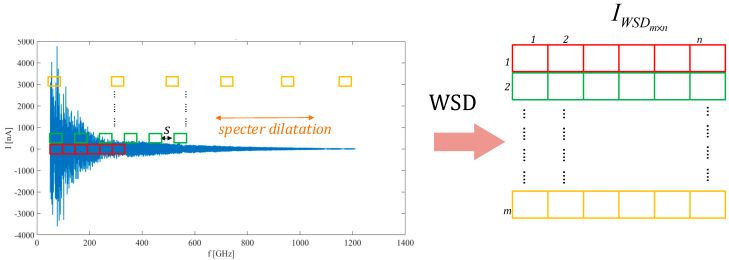
The windowing spectrum dilation (WSD) transformation technique.

**Figure 9 sensors-21-04709-f009:**
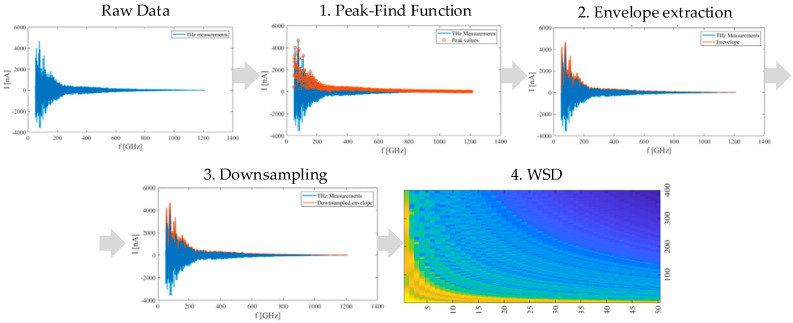
Preprocessing sequences; peak-find function, envelope extraction, downsampling, and WSD algorithm.

**Figure 10 sensors-21-04709-f010:**
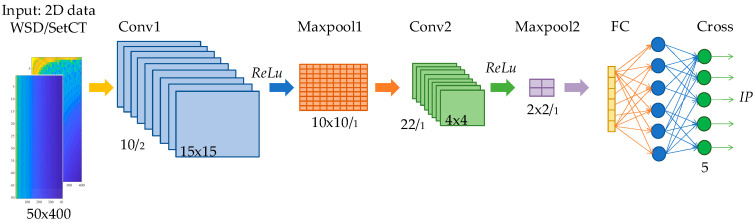
Convolutional neural network (CNN) structure for full-resolution 2D data.

**Figure 11 sensors-21-04709-f011:**
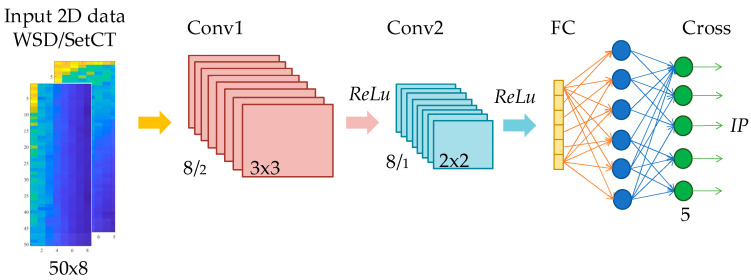
CNN structure for low-resolution 2D data.

**Figure 12 sensors-21-04709-f012:**
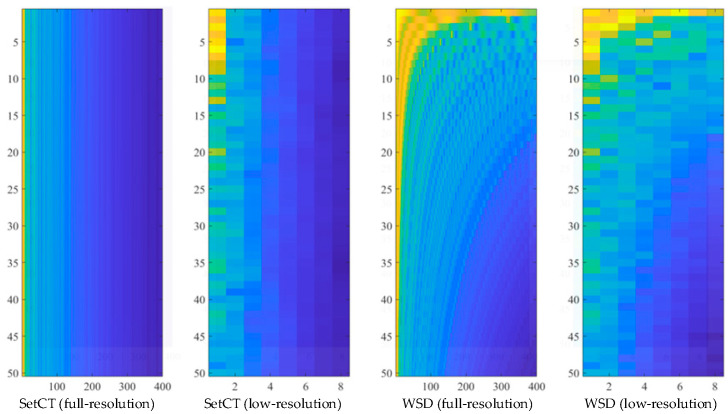
SetCT and WSD 2D transformation.

**Figure 13 sensors-21-04709-f013:**
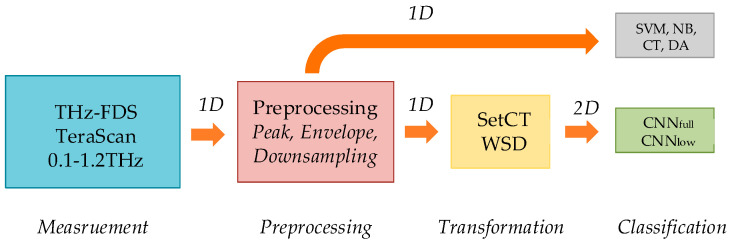
The experimental procedure.

**Figure 14 sensors-21-04709-f014:**
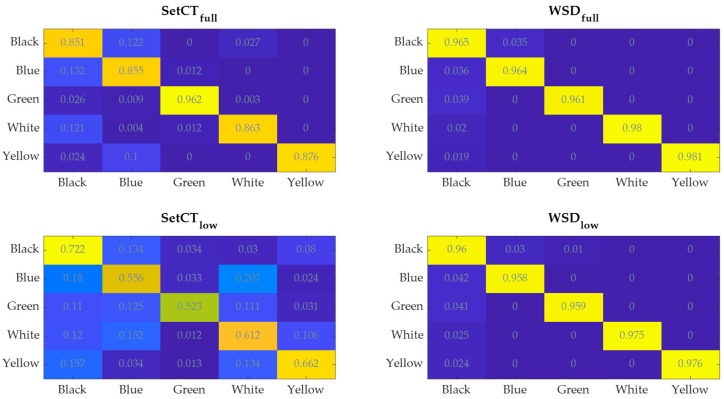
Confusion matrix of SetCT_high_, SetCT_low_ WSD_high,_ and WSD_low_.

**Figure 15 sensors-21-04709-f015:**
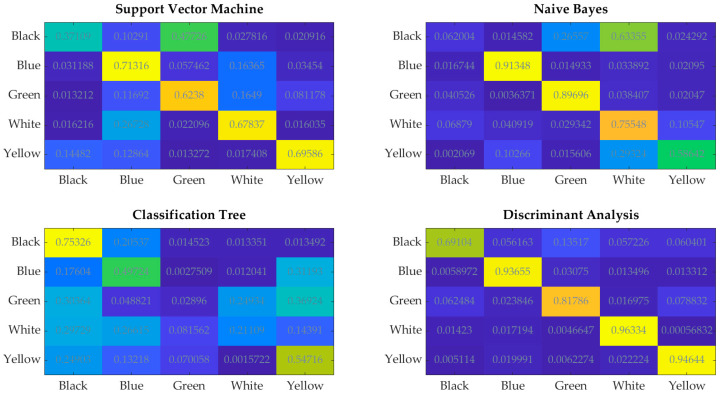
Confusion matrix of support vector machine (SVM), naive Bayes (NB), classification tree (CT), and discriminant analysis (DA) classification algorithms.

**Table 1 sensors-21-04709-t001:** CNN’s learning parameters and results.

	$\mu_{l}$	$t_{l}\left(h\right)$	α	$epoch_{max}$
**SetCT_full_**	0.965	5.40	0.001	1500
**SWD_full_**	0.972	4.10	0.001	1500
**SetCT_low_**	0.85	0.35	0.003	350
**SWD_low_**	0.968	0.15	0.003	350

**Table 2 sensors-21-04709-t002:** Performance evaluation of the THz-FDS classification of a black sample.

Black	$t_{t}\left(ms\right)$	$\mu_{t}$	$\sigma_{t}$	$\mu_{tmax}$	$\mu_{tmin}$
**SetCT_full_**	241.5	0.951	0.054	1	0.35
**SWD_full_**	242.7	0.991	0.039	1	0.41
**SetCT_low_**	1.42	0.722	0.250	0.933	0.23
**SWD_low_**	1.65	0.985	0.041	1	0.37

**Table 3 sensors-21-04709-t003:** Performance evaluation of the THz-FDS classification of a blue sample.

Blue	$t_{t}\left(ms\right)$	$\mu_{t}$	$\sigma_{t}$	$\mu_{tmax}$	$\mu_{tmin}$
**SetCT_full_**	240.1	0.955	0.037	1	0.38
**SWD_full_**	241.2	0.984	0.036	1	0.37
**SetCT_low_**	1.71	0.756	0.231	0.93	0.25
**SWD_low_**	1.66	0.979	0.039	1	0.33

**Table 4 sensors-21-04709-t004:** Performance evaluation of the THz-FDS classification of a green sample.

Green	$t_{t}\left(ms\right)$	$\mu_{t}$	$\sigma_{t}$	$\mu_{tmax}$	$\mu_{tmin}$
**SetCT_full_**	242.2	0.962	0.031	1	0.391
**SWD_full_**	244.1	0.977	0.033	1	0.392
**SetCT_low_**	1.52	0.723	0.251	0.943	0.311
**SWD_low_**	1.71	0.976	0.029	1	0.381

**Table 5 sensors-21-04709-t005:** Performance evaluation of the THz-FDS classification of a white sample.

White	$t_{t}\left(ms\right)$	$\mu_{t}$	$\sigma_{t}$	$\mu_{tmax}$	$\mu_{tmin}$
**SetCT_full_**	243.1	0.963	0.028	1	0.423
**SWD_full_**	245.1	0.984	0.023	1	0.457
**SetCT_low_**	1.22	0.812	0.128	0.935	0.341
**SWD_low_**	1.31	0.981	0.032	1	0.421

**Table 6 sensors-21-04709-t006:** Performance evaluation of the THz-FDS classification of a yellow sample.

Yellow	$t_{t}\left(ms\right)$	$\mu_{t}$	$\sigma_{t}$	$\mu_{tmax}$	$\mu_{tmin}$
**SetCT_full_**	242.23	0.979	0.023	1	0.533
**SWD_full_**	242.72	0.986	0.024	1	0.512
**SetCT_low_**	1.71	0.762	0.169	0.919	0.216
**SWD_low_**	1.45	0.984	0.025	1	0.491

**Table 7 sensors-21-04709-t007:** The comparison of classification algorithms’ performance.

Algorithm	$t_{t}\left(ms\right)$	$\mu_{a}$	$\sigma_{t}$	$\mu_{tmax}$	$\mu_{tmin}$
**SetCT_full_**	242.23	0.962	0.023	1	0.533
**SWD_full_**	242.72	0.984	0.024	1	0.512
**SetCT_low_**	1.71	0.755	0.169	0.919	0.216
**SWD_low_**	1.45	0.981	0.025	1	0.491
**SVM**	1.17	0.73	0.23	0.851	0.123
**NB**	1.28	0.74	0.25	0.898	0.231
**CT**	1.21	0.51	0.31	0.71	0.02
**DA**	1.24	0.89	0.102	1	0.49

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
