# Peer review of "Automated Inorganic Pigment Classification in Plastic Material Using Terahertz Spectroscopy"

_sensors, 2021, doi:10.3390/s21144709_

Round 1

Reviewer 1 Report

From the viewpoint of THz-spectroscopy, the revised manuscript is still unclear and in a diffuse style.  Almost only the details of the processing techniques are emphasized and I do not well understand the motivation, necessity, result or fruit of this study.  I do not understand why the authors did not employ THz-TDS data but THz-FDS data, which is much more difficult to be analyzed than the THz-TDS data.  I do not understand why the authors utilized the complex processing techniques on the complex experimental data (THz-FDS).

Reviewer 2 Report

(1) this paper researches a automated inorganic pigment classification in plastic material using terahertz spectroscopy, the research idea is good and the description is clear. However, this paper does not reflect the research on sensors, so it is suggested to reflect it in the paper.

(2)It is suggested that the author give the design method of the test system.The paper reflects as far as possible the research of the sensor or the improvement of the sensor, etc.

Round 2

Reviewer 1 Report

The authors have responded to only a part of my comments.  I am afraid that I cannot expect sufficient improvement within a couple of exchanges.  However, they have responded the most basic comment and the term of this review is only three days.  Then I am not against the acceptance.

There are indentations in lines just after the equations.  They should be removed.

Author Response

The answer is in the attachment. 

This manuscript is a resubmission of an earlier submission. The following is a list of the peer review reports and author responses from that submission.

Round 1

Reviewer 1 Report

The paper introduces a convolution neural network as a classifier and a novel preprocessing THz raw data methodology. The proposed preprocessing method covers four steps, where the first three are peak detection, envelope extraction, and downsampling algorithms. The last step, the Windowing with Spectrum Dilatation, is introduced. The major concern of mine is that the novelty of the work is insufficient. It seems that the paper simply use a CNN for ThZ image classification, there are is no contribution on the network design, as well as new formulation of the loss. Moreover, there lacks baselines for comparsion. 

Reviewer 3 Report

The paper from Sarjas et al. describes a convolutional neural network analysis of inorganic pigments from THz spectroscopy data. 

The first 6 pages of the paper provide unnecessary background information on THz spectroscopy, which would be more appropriate for a thesis, rather than for a scientific article. Also the following description of the data treatment is excessively long, so that the description of the experiment starts at page 11 only. The final results are provided in a series of tables, which would be acceptable for a technical report, not for a paper. The discussion of the results is done in less than one page, mainly providing a comparison between different data treatments. After reading the paper it is not clear to me why one should prefer neural networks with respect to other classification schemes.

I found the manuscript badly structured, of average quality and poor english.